# SLIDE—Novel Approach to Apocrine Sweat Sampling for Lipid Profiling in Healthy Individuals

**DOI:** 10.3390/ijms22158054

**Published:** 2021-07-28

**Authors:** Aleš Kvasnička, David Friedecký, Alena Tichá, Radomír Hyšpler, Hana Janečková, Radana Brumarová, Lukáš Najdekr, Zdeněk Zadák

**Affiliations:** 1Faculty of Medicine and Dentistry, Palacký University Olomouc, 779 00 Olomouc, Czech Republic; ales.kvasnicka01@upol.cz (A.K.); radana.karlikova@gmail.com (R.B.); lukas.najdekr@gmail.com (L.N.); 2Laboratory for Inherited Metabolic Disorders, Department of Clinical Chemistry, University Hospital Olomouc, 779 00 Olomouc, Czech Republic; janeckovah@gmail.com; 3Department of Clinical Biochemistry and Diagnostics and Osteocenter, University Hospital Hradec Králové, Sokolská 581, 500 05 Hradec Králové, Czech Republic; alena.ticha@fnhk.cz (A.T.); radomir.hyspler@fnhk.cz (R.H.); 4Department of Research and Development, University Hospital Hradec Králové, Sokolská 581, 500 05 Hradec Králové, Czech Republic; zdenek.zadak@fnhk.cz

**Keywords:** apocrine sweat, lipidomics, mass spectrometry, microsampling, profiling

## Abstract

We designed a concept of 3D-printed attachment with porous glass filter disks—SLIDE (Sweat sampLIng DevicE) for easy sampling of apocrine sweat. By applying advanced mass spectrometry coupled with the liquid chromatography technique, the complex lipid profiles were measured to evaluate the reproducibility and robustness of this novel approach. Moreover, our in-depth statistical evaluation of the data provided an insight into the potential use of apocrine sweat as a novel and diagnostically relevant biofluid for clinical analyses. Data transformation using probabilistic quotient normalization (PQN) significantly improved the analytical characteristics and overcame the ‘sample dilution issue’ of the sampling. The lipidomic content of apocrine sweat from healthy subjects was described in terms of identification and quantitation. A total of 240 lipids across 15 classes were identified. The lipid concentrations varied from 10^−10^ to 10^−4^ mol/L. The most numerous class of lipids were ceramides (*n* = 61), while the free fatty acids were the most abundant ones (average concentrations of 10^−5^ mol/L). The main advantages of apocrine sweat microsampling include: (a) the non-invasiveness of the procedure and (b) the unique feature of apocrine sweat, reflecting metabolome and lipidome of the intracellular space and plasmatic membranes. The SLIDE application as a sampling technique of apocrine sweat brings a promising alternative, including various possibilities in modern clinical practice.

## 1. Introduction

The entire sweat gland system represents 1.6–5 million apocrine and eccrine glands distributed throughout the body surface [1] as an integral part of one of the largest organs—the skin [2]. The eccrine sweat glands are the most numerous, spread over almost the entire surface of the body. They are responsible for the largest volume of the sweat secretion, which fulfills the thermoregulatory function of the body [3]. The eccrine glands develop until the age of 2–3 years—when they become active, and their quantity stays constant throughout life. The development of apocrine sweat glands begins at birth, but they become active later during the stimulation by androgens or estrogens—associated with puberty. After a period of sexual maturity, their activity gradually decreases with age. The main concentration of apocrine glands is on the hairy parts of the body (in the axillae, mammary areolae, and periumbilical and genital areas). The armpit contains the most significant number of apocrine glands on the entire surface of human skin [2]. Although a mixed-type apoeccrine gland has been described [4], other authors have not confirmed its presence [5]. Therefore, only two types of sweat glands are usually mentioned in the literature.

To characterize sweat glands by their secretory mechanism, they can be divided into three main groups: holocrine, merocrine (also called eccrine) and apocrine (Figure 1). (1) The holocrine secretion is defined as a complete disintegration of a cell into a secretion (Figure 1c), which is typical for sebaceous glands. They are also commonly found near the hair follicles and other sweat glands. However, they cannot be considered sweat glands, as they form the lipid-rich secretion by breaking its whole cell down. On the other hand, the eccrine and apocrine glands secrete in the true sense of the word, and do not undergo complete disintegration. (2) Merocrine secretion is based on the formation of vacuoles from Golgi’s apparatus and the cell membrane. Formed vacuoles and secretory granules, containing mainly water-soluble metabolites, ions and proteins, leave the cell via the process of exocytosis without disrupting it (Figure 1b). (3) Apocrine secretion is characterized by separating a part of the cell containing cytoplasm and cell membrane (Figure 1a). Unlike holocrine secretion, which occurs after death and the breakdown of the entire cell, apocrine secretion is formed from a living cell. It delivers part of the cell’s cytoplasm and cell membrane, containing intracellular fragments, to the outlet of the apocrine gland [6]. The released part of the cell grows back, and the cell continues to function. Therefore, its utility lies in the advantage of apocrine secretion that reflects the intracellular space’s metabolic ratios. On the other hand, routine clinical materials such as urine or blood plasma (or serum) mainly reflect the metabolic situation in the extracellular space, and hence only partially and indirectly the intracellular and membrane metabolic processes [6]. Although the mechanism of apocrine sweat formation and function has been studied for a long time, it has not yet been completely elucidated [7].

The easier acquisition of eccrine sweat, which can be supported thermally or by physical activity, led to its better chemical characterization than apocrine sweat. Still, due to compositional similarities, both will be discussed together. The sweat consists of three main parts: (1) water which makes up the majority of sweat (almost 99%); (2) ions—Na^+^, Cl^−^, Ca^2+^, K^+^, Mg^2+^, PO_4_^3−^, NH_4_^+^ and others, including ionic species, and (3) organic molecules—amino acids, hydroxy/keto acids, free fatty acids, saccharides, urea, creatinine, uric acid, vitamins, volatile compounds and intact/metabolized exogenous chemicals and pharmaceuticals [8]. Sweat is also often attributed to the function of removing metabolic wastes through perspiration, but so far, this function seems to be relatively negligible and mostly reserved for the kidneys [9]. However, it appears that various exogenous substances and toxins such as pharmaceuticals [10], drugs [11], toxic metals [12] or other organic substances [13] are transferred into a sweat (and the concentrations in sweat are often higher than that of blood and/or urine), which could be used for monitoring or screening purposes.

Additionally, apocrine sweat represents the chemical nature of body odor, caused by the microbial (mainly the *Corynebacterium* genus) transformation of steroid molecules, volatile and other organic fatty acids and lipids secreted on the skin surface by sweat glands [14]. This makes apocrine sweat a sensitive indicator for specific odor modulating substances, e.g., in some spices, and it is also possible to observe these compounds metabolically transformed [15]. To clarify the composition and origin of the apocrine secretion (not only sweat but also, for example, saliva, tears, or milk), various proteomic analyses were performed, describing cytoskeletal, membranous, microsomal, ribosomal, mitochondrial, and even, nuclear and nucleolar proteins in these biofluids [6,16]. In terms of similarity of the proteomic composition of biofluids produced by apocrine glands (milk, sweat, tears, cerumen, saliva and cerebrospinal fluid), apocrine fluids contain between 30 to 65% identical proteins regardless of their anatomical origin [6]. Furthermore, the proteomes of all the human apocrine fluids compared to plasma or serum revealed that 38 (cerebrospinal fluid) to 91% (milk) of the entries were identical to plasma proteins [17]. This fact points to a link between the apocrine secretion (on the surface of the body) and the plasma, representing the situation in the blood circulation.

The sampling of apocrine sweat possesses a challenge as the secretion volume is very small, equivalent to a few microliters in one sample taken [18]. Nevertheless, under certain circumstances (heat, exercise, influence of pain, topical application), apocrine sweat is diluted by eccrine sweat. Among other properties, apocrine sweat differs from eccrine in its pH. Apocrine sweat (pH 5.0–6.5) has an average pH approximately 0.5 units higher than eccrine sweat (pH 4.0–6.0). Using the pH value, the ratio between apocrine and eccrine sweat in the sample can be approximately estimated [18]. Another technique to distinguish between the two types of sweat is fluorescence under ultraviolet light, as only apocrine sweat shows fluorescence [18]. A reliable way to collect apocrine sweat, which is not diluted by eccrine sweat, is to expel the contents of the apocrine glands using external compression, vacuum or capillary forces [4,18]. These techniques produce extra apocrine sweat without significant contamination of the eccrine component (if the subject’s physiology is not further influenced by heat and/or exercise). A major issue in the collection of apocrine sweat from armpit apocrine glands is the frequent use of antiperspirants (which leads to obstruction of the apocrine gland outlets, as well as damage to the apocrine gland itself) and difficult reproducibility of sampling (which is caused by an individual amount of secretion and a non-specific volume of the sample taken) [19].

Future trends in clinical diagnostics are moving toward small sample volumes, high sensitivity of analytical methods, non-invasive collection techniques and a shift toward alternative biofluids. Although apocrine sweat is not yet used as a routine clinical material, it offers potential applications in screening (drugs, pharmaceuticals), monitoring (diseases, therapeutic drugs) or diagnostics [20,21]. For clinical applications, apocrine sweat has the following advantages and significance:It is a secretion that reflects the contents of the cellular cytoplasm (including cellular fragments).It reflects the components contained in the cell membrane of apocrine gland secretory cells.The components correspond to the cytoplasm composition and the cell membrane of a living cell (as opposed to a holocrine secretion, which is composed of the remnants of dead cells).The lipid nature of apocrine secretion may be used in the future alongside diagnostic targets to identify/quantify lipid xenobiotics, lipophilic pharmaceuticals, and lipophilic narcotic drugs.

Apocrine sweat is still an incompletely studied biological fluid in metabolic and lipid composition, as comprehensive studies are lacking. The future use of apocrine sweat in clinical practice will become apparent when more studies comprehensively characterize this material. However, it requires the availability of highly sensitive microscale analytical techniques as only a few microliters of a sample can be acquired. The current study aimed to develop a robust and straightforward apocrine sweat sampling technique and verify the reproducibility of such a procedure. The sampling technique was evaluated by determining the lipidomic profile in healthy subjects following a standard sampling protocol. Subsequently, the data obtained were statistically evaluated further to understand the variability and behavior of apocrine sweat.

## 2. Results

### 2.1. Apocrine Sweat Microsampling Technique

SLIDE consists of a commercially available porous glass filter disk and a custom 3D-printed attachment (Figure 2a). These main parts are assembled together as shown in Figure 2c, where the 3D-printed attachment serves as a handle and the porous glass filter disk is used to directly swipe the desired part of the skin of the armpit (for further explanation of the sampling technique see Section 4.2).

In practical experience, the sampling procedure proved to be noninvasive, painless and easy to perform (approximately 5 min, including informing the participant and the collection itself). Furthermore, volunteers did not experience any discomfort, such as dermatographic urticaria or skin irritation from the 80% isopropanol solution. In this regard, the similarity of this solution to a widely used preinjection alcohol swab should be mentioned [22]. Immediately after sampling the volunteers were able to commence their standard daily activities.

### 2.2. Pseudotargeted Lipidomic Analysis

The pseudotargeted approach consists of calculating ion pairs (multiple reaction monitoring transitions—MRM) corresponding to theoretical lipids in lipid classes possibly present in the sample. The selection of relevant MRM transitions is carried out first. During the optimization step, three analytical methods (two in positive and one in negative ion mode) consisting of approximately 3800 different MRM transitions were used to adjust the final lipidomic method. Based on the analysis of quality control samples (QC, criteria specified in Section 4.5), the final list of 240 detected lipids and the corresponding MRM transitions was created to measure all samples (Appendix A). The identified lipids are belonging to 15 lipid classes and subclasses as follows: cholesteryl esters (CE), ceramides (Cer), hexosylceramides (HexCer); lysophosphatidylcholines (LPC); phosphatidylcholines (PC); plasmalogen phosphatidylcholines (PCO); lysophosphatidylethanolamines (LPE); phosphatidylethanolamines (PE); plasmalogen phosphatidylethanolamines (PEO); phosphatidylinositols (PI); phosphatidylserines (PS); sphingomyelins (SM); diacylglycerols (DG); triacylglycerols (TG) and free fatty acids (FA). Expanded stacked chromatograms showing each lipid class separately in more detail are provided in the Appendix A.

To test the analytical specificity and robustness to interferences of the final lipidomic method, the extracts from the 3D-filament were analyzed. After incubation of the plastic attachment in 80% isopropanol solution, the interferences of contaminating substances were marginal. Only one peak corresponding to MRM of Cer d20:1/18:0 was observed in the plastic extract (Appendix A). Several FA species were also detected in the process blanks (FA16:0 and FA18:0), as common residues of organic solvents. A post-column infusion experiment was carried out to ensure that no significant matrix effects occur. The total ion chromatogram (TIC of isotopically labeled standards) baseline oscillated at intensities of 2 × 10^7^ for both QC sample and process blank in the positive ion mode (Appendix A), and most of the matrix-induced deviations to higher intensities showed narrow peaks and occurred in the elution periods between the retention windows of individual lipid classes. The injection volume experiment showed a linear response (R^2^ > 0.95) for the majority of all lipids (206/240, Appendix A).

When comparing the extracted ion chromatograms (in MRM mode), significant differences in lipid profiles were found not only in the intensities of the detected lipids (y-axis) but moreover in their different compositions between individuals (Figure 3). Massive changes are evident in the glycerophospholipid and sphingolipid regions of the chromatogram (Figure 3), indicating the high biological variability of this material. Therefore, a sophisticated data processing strategy is required.

### 2.3. Importance of Sweat Lipidome Data Transformation

To overcome an issue of different volumes of samples (no standardization available), a data transformation based on partial quotient normalization combined with the natural logarithm (lnPQN) was applied and compared with the commonly used natural logarithm (ln). The lnPQN is based on the calculation of medians across the samples (observations) and lipids (variables) and their relation to each lipid [23]. In recent years, the lnPQN transformation has found its indispensable place in metabolomics/lipidomics experiments and provides promising results for many sample types where other normalization methods fail (more details are provided in Section 3.3) [24].

To evaluate the data transformation methods, univariate and multivariate statistical methods were applied. For univariate statistics, intra-individual variability (CVi), group variability (CVg), and variability for quality control samples (CVqc) were calculated for each lipid. A dramatic decrease to approximately half the CVi (67 vs. 36%) and CVg (112 vs. 72%) values were observed when using lnPQN compared to ln while keeping the same CVqc (7.7 vs. 7.9%), (Figure 4, Appendix A). This shows the importance of using the lnPQN transformation to reduce variability either within or between individuals.

Multivariate statistical approaches offer a unique view of the general behavior of samples. Unsupervised hierarchical cluster analysis (HCA) and principal component analysis (PCA) show similar trends as the results from univariate statistics. In HCA, the clustering of only two individuals (participants C and I) can be seen in ln transformed data compared to lnPQN transformed data where almost all individuals are grouped (except participant D), (Figure 5, Appendix A). Moreover, the y-axis representing the importance of differences is more than two times higher in the ln transformed data.

The score plot of PCA in Figure 6 projects all lipid variables describing the samples in two dimensions. The grouping of samples based on an individual could be more clearly observed in the case of lnPQN compared to ln (Figure 6, Appendix A). Explained variances of the PCA model according to scores 1 and 2 were 56.6/28.9% and 26.0/11.8%, respectively. The twice higher variance in the first score t [1] for ln transformed data compared to lnPQN explains the significantly higher dispersion and variability of the data. Additionally, the strong grouping of quality control samples placed close to the center point (gray color) confirms the overall quality of the lipidomic experiment. Participant I, whose samples lay farthest from the others on PCA, was on average 60 years older than the other subjects in our study. This indicates that age could be an important factor reflecting changes in apocrine sweat composition, and additional studies with larger cohorts (homogeneous categories by age and sex of participants) will be needed to precisely describe and explain the impact of age.

### 2.4. Intraindividual and Group Variability of Lipids

The variabilities of each lipid class were calculated. The summarized average variabilities for detected lipid classes vary 26–64% and 51–119% for CVi and CVg (Figure 7), respectively. The higher variability can be seen in groups with lower abundances e.g., in PI or PS classes. Conversely, the lowest variabilities are present in highly abundant lipid classes such as FA, TG, and Cer.

To visualize the distribution of CVi and CVg for all detected compounds, lipid networks were constructed in Cytoscape software where each lipid class forms a cluster, with a description of the individual lipids assigned to it (Figure 8 and Figure 9) [25]. Lipids were annotated according to the generally accepted lipid nomenclature [26] and the level of annotation (e.g., acyl-chain composition, or sn-1/sn-2 position) matched the resolving possibilities of the analytical method (described in more detail in Section 4.5). In addition to the trends observed in Figure 7, discrepancies within the classes were seen. In the TG class, which showed low average variability of approximately 30% in the whole group, only TGs of 54 carbons and 2–6 double bonds showed more than double CVi compared to the rest of the class. Similarly, the CVg for a few Cer species almost doubled from the mean class CVg (Figure 9). These deviations can be explained by the possible variability of origin of lipids (apocrine, eccrine, sebaceous), but also, for example, by the concentration or stability of these lipids.

### 2.5. Quantitation of Lipid Classes

Concentrations of lipids in biological samples usually differ by many orders of magnitude. Similarly, in sweat extract a large dispersion of the concentration levels between and within lipid classes can be observed. The most abundant FA, CE, and DG classes lie in the median of 10^−5^ mol/L compared to low abundant LPE, LPC, PEO and PCO with a median of 10^−9^ mol/L (Figure 10). Finally, the overall detectable dispersion of concentration levels is more than five orders of magnitude (from 10^−10^ to 10^−4^ mol/L). Moreover, the numbers of detected lipids significantly differed across the classes from only a few in the PI class up to dozens in Cer and TG classes.

If the concentrations of all lipids in each class are summed, a large representation of FA (83.8%) followed by DG (7.9%), CE (4.4%), TG (2.1%) can be observed (Figure 11a). Sphingolipids (SP) and glycerophospholipids (GP) collectively represented 1.9% of the rest of the lipids, and ceramides (1.3%) and sphingomyelins (0.2%) were making up a majority of these subclasses (Figure 11b).

### 2.6. Correlation of Lipids in Apocrine Sweat

To overview the relationships between lipid classes in sweat, Pearson’s linear correlation was performed and the results visualized in a heatmap (Figure 12). Based on an experimental setting (*n* = 60 samples), correlation coefficients 0.26 and higher were calculated as statistically significant, where medium and strong correlations were higher than 0.5 and 0.7, respectively. Systematic trends can be observed as lipids in one class generally behave similarly compared to another lipid class (but interclass trends can be also observed). For example, ceramides show a negative correlation with most other lipid classes except free fatty acids to which they correlate positively, and triacylglycerols with both positive and negative correlation regions on the heatmap. A positive correlation among all glycerophospholipids can be observed except for the negatively correlated PEO and PS classes.

### 2.7. Armpit Side-Specific Differences

In addition to the basic description of the lipid composition of apocrine sweat, the differences of sampling between the left and right armpits were studied. The visualization of all lipids categorized by lipid classes was performed using Cytoscape software (Figure 13). Finally, samples taken on the same day from each person were statistically evaluated using paired *t*-test and fold-change. When Benjamini-Hochberg correction (false discovery rate adjusted to 0.25) was applied, none of any of the lipids was evaluated as statistically significant. Despite systematic trends of increased ceramides, diacylglycerols, and partly decreased selected glycerophospholipids (PC, PCO, PE, PEO) can be seen, the relative changes are lower than 15% for the majority of lipids.

## 3. Discussion

### 3.1. Novel Approach for Sweat Sampling

In the study presented here, we focused on characterizing apocrine sweat as a potentially clinically relevant biofluid, from multiple points of view. First, we have developed and described a simple, easy-to-use sampling technique, using a custom 3D-printed attachment and a commercially available porous glass filter disk. This concept not only offers an opportunity for other researchers and clinicians to adopt our technique for further studies, but it also brings a unifying aspect to apocrine sweat sampling, as the 3D blueprint can be easily shared and modified for further development. We have also described a standardized protocol with recommendations for consistent sampling. The usage of antiperspirants and other cosmetics with underarm application needs to be avoided before the sampling of apocrine sweat as it can lead to obstruction of apocrine sweat glands [19]. Due to the physiology of the human secretory glands, it is not possible to obtain pure apocrine sweat because in the axillary region, in addition to the high density of apocrine glands, eccrine glands and sebaceous glands (in the hair follicles) are also present [1]. It should also be noted that participants’ physiology was not altered by heat or physical activity (stimulating sweat production in other glands), and therefore an assumption was taken that the majority of sampled material is of apocrine gland origin and minor parts are of eccrine, sebaceous, or even microbial sources. To avoid this contamination as much as possible, the armpit was carefully cleaned with distilled water prior to the sampling. A perfectly adequate name for the material studied in this work is therefore armpit sweat (the main component of which is apocrine sweat). With defined and optimized the sampling technique, we proceeded to a detailed description of the material itself.

### 3.2. Lipidomic Methodology

Lipidomic analysis is increasingly showing its potential in the field of clinical diagnostics as well as offering unique insights into the biochemistry and biophysics of biological fluids and tissues [27]. Apocrine sweat is an ideal material for lipidomic analysis as it contains not only intracellular metabolites but also lipid bilayers, which are released by apocrine glands [6]. The absolute quantitative LC-MS analysis of lipids can be achieved only using HILIC separation, where all lipids are separated based on their polar head groups and not the side chains. Such mechanism is providing identical ionization conditions and matrix effects for each lipid class [28]. On the other hand, the complementary reversed-phase mode (RP) provides separation based on acyl-chain length and number of double bonds, which offers a resolution of isomeric lipids. Moreover, using lipid pattern plots allows easy deciphering of misidentifications. Recently, it has been shown that RP and HILIC-based LC-MS analyses of lipids yield equal quantitative results for many lipid classes [29]. As our goal was to comprehensively describe the lipid composition of the sampled sweat, as well as to quantify these lipids, the RPLC system was used. However, it needs to be addressed that for absolute quantitation multiple lipid standards within each lipid class would have to be used.

### 3.3. Data Processing Solution to the Problem of the Variability

Descriptive statistical evaluation of the studied material is necessary for its use in clinical applications. As sweat had already been shown to be a highly variable material as regards water content and concentration of ions [1,20,30] and considering the fact that our approach did not account for differences in the volume of the sample, a robust normalization strategy had to be chosen. For the analysis of urine, the creatinine concentration is commonly used as a normalization parameter of the dilution, but it still does not solve all problems when performing metabolomic urinary analysis [31]. The Probabilistic Quotient normalization (PQN) offers a solution to the dilution-induced sample inconsistencies as it transforms the metabolomics data according to an overall estimation of the most probable dilution [23]. Although this normalization was first applied to NMR data, it is now becoming increasingly popular in metabolomics and lipidomics as it outperforms other normalization strategies [24]. Our goal was to obtain data of as low non-biologically induced variability as possible while at the same time preserving information about individual biological variability. Finally, the application of lnPQN transformation was able to reduce CVi and CVg to approximately half that using natural logarithm alone, while the variability of QC samples remained unchanged, indicating the selectivity of the lnPQN approach in avoiding an overestimated normalization of the data.

The general principle and usefulness of the lnPQN were additionally proven by multivariate statistical analysis, particularly in the hierarchical cluster analysis, where 9 out of 10 individual participants were clustered, while an effect of the different day or side of sampling was not seen. Further insight into biological variability was offered by the results from PCA, in which all individual samples clustered close together, but 3 participants (C, I, F) lay further away from all other samples. This outcome highlights one imperfection of our study, namely the uneven representation of ages in our small cohort. These results are also in compliance with larger cohort studies investigating correlations between plasma lipids, age, and sex of participants, where intra-/interclass trends in lipids showed a good correlation with age [32].

### 3.4. Description of the Sweat Lipidome

In terms of the lipid concentration in apocrine sweat, FA dominated, followed by glycerolipids (TG and DG) and then CE and phospholipids (SP and GP). Takemura et al. [33] discovered that the technique of sweat collection significantly affects its composition, and that the FA content of physically scraped sweat can be up to 100-fold higher than that of clean sweat (collected passively without scraping). As far as we know this is the first work comprehensively describing the apocrine sweat lipidome including estimation of lipid concentrations. We compared our results with those from other studies of sweat from various origins. Peter et al. [34] subjected mechanically collected eccrine sweat to quantitative FA analysis and found that FA concentration oscillated between 0.1–5.47 μg/mL, which is approximately one order of magnitude below that in our measurements (in which FA ranged from 0.03–44.4 μg/mL). The difference can be explained by the different biological origin (eccrine gland) of the sweat in the aforementioned study and the higher extraction efficiency of our sampling procedure (due to pre-soaking the disk in 80% IPA). The high concentration of lipids (specifically FA) is consistent with the assumption that apocrine sweat is a lipid-rich material and, unlike eccrine sweat, cannot be completely evaporated [18]. As the release of apocrine sweat is promoted by physical stimulation, we chose the scraping approach for its collection, which can lead to an increased content of FA in the final sample, similarly to the presence of sterols and ceramides of sebaceous and epidermal origin. The heterogeneous lipid composition of the sample was also reflected in the number of individual lipids (in parentheses), where Cer (*n* = 61) were the most represented, followed by TG (*n* = 37), SM (*n* = 23), PC (*n* = 20), DG (*n* = 17) and FA (*n* = 17). At first glance the large number of ceramides of structurally heterogeneous character (various long-chain bases: 16:0, 16:1, 18:0, 18:1, 20:1) may be striking; however, they are found abundantly as constituents of the cutaneous lipidome, where their function (water permeability barrier of the skin) and their role in pathobiochemical processes (e.g., atopic dermatitis or psoriasis caused by disruption of the permeability function or inflammatory disbalance of lipids in the epidermis) have been well documented [35,36,37].

To understand the biochemistry and biological behavior of a particular material, it is helpful to focus on the trends within and between groups of studied biomolecules in a particular biofluid or tissue. By constructing a correlation map, we could easily trace the relationship trends between lipid classes. We could also observe that one lipid class generally behaves in a consistent manner compared to another class, which is commonly found in studies of plasma lipidome [38]. Glycerolipids (TG and DG) in apocrine sweat did not show a similar correlation pattern and were frequently showing opposite trends compared to each other. In contrast, in plasma these classes usually correlate in a similar pattern together [39]. Interestingly, all glycerophospholipids showed similar trends in correlations between and within lipid classes (except for the correlation of PS and PI to PEO). On the other hand, in common with other researchers, we have also found correlation trends in the classes themselves and in some cases even individual lipids with opposite behavior to the rest of the class (indicating different origin, function, or regulation) [39].

To fully describe the studied material, differences between the sides of sampling were also evaluated. Although systematic trends in lipid class levels were observed between the left and right sides of sampling, these changes were considered nonsignificant after the application of *p*-value correction (Benjamini-Hochberg). Furthermore, since these differences for the majority of lipids were only up to +/−15%, this contribution was minor compared to the interindividual variability of individual lipid classes (which are mostly greater than 50%). On the other hand, these differences could be of significant importance for diseases affecting only one of an organ pair (e.g., breast cancer), or they could also reflect the physiological dominance of the right or left hand and/or lymph nodes [40]. However, further studies will be needed to clarify these hypotheses.

### 3.5. Future Development

The results presented above show the potential of apocrine sweat sampling and its analysis by advanced mass spectrometry coupled with liquid chromatography, as a promising noninvasive alternative to established methods in routine clinical analysis. Based on the results of our research and also from other studies, it is likely that new unsuspected applications for various microsampling techniques will be emerging for both diagnostic and therapeutic purposes [41]. Given the large number of different techniques already described for the collection of apocrine sweat (which usually involve stimulation by heat, physical activity, pain, or the intradermal injection of adrenaline), it would be useful to compare them with each other and with the SLIDE technique in the following studies. Large-scale studies will be necessary to define the influence of age, gender, BMI, and other factors on the lipidome of apocrine sweat. Due to the low cost of the sampling method itself, the potential future application can be also seen in screening programs. This study was conducted to improve characterization of the apocrine sweat lipidome, as other pilot experiments have already shown the usefulness of sweat analysis for (1) drug screening [11], (2) detection of early epidermal immunomodulation disruption in atopic dermatitis [36], (3) glucose monitoring in diabetic patients [42] and (4) detection and measurement of pharmaceuticals [10]. This work was carried out because a reliable sampling technique and analysis of the composition of apocrine sweat in healthy individuals is a necessary basis for the subsequent use of this biofluid in modern clinical diagnostics.

## 4. Materials and Methods

### 4.1. Chemicals and Reagents

Acetonitrile (ACN), isopropanol (IPA), water, and ammonium acetate (all LC/MS grade) were purchased from Sigma-Aldrich (St. Louis, MO, USA). SPLASH^®^ LIPIDOMIX^®^ Mass Spec Standard mixture and ceramide (d18:1-d7/15:0) were purchased from Avanti Polar Lipids (Alabaster, AL, USA). Arachidonic acid-d8 was acquired from Cayman Chemical Company (Ann Arbor, MI, USA).

### 4.2. D-Printed Attachment

The attachment was constructed in 3D Builder (version 18.0.1931.0, Microsoft Corporation, Redmond, WA, USA). The external dimensions of the attachment were 41 mm in length and 14 mm in radius (Figure 2). A 1.75 mm diameter, sapphire-colored PLA filament (Smart Materials 3D, Alcalá la Real, Spain) was used for printing. The attachment was printed on DeltiX 3D (Trilab, Brno, Czech Republic) under standard conditions typical for PLA filaments (extruder and bed temperatures: 210/55 °C; extrusion width: 0.4 mm; layer thickness: 0.2 mm; speed: fast). The model in 3mf and stl format is available in the Appendix A.

### 4.3. Sampling Technique

Apocrine sweat was collected in gloves using a custom 3D-printed attachment with porous glass filter disk (porosity S1, 10 mm diameter, Figure 1) purchased from Winzer Laborglastechnik (Wertheim, Germany). We have called this system SLIDE, which is an acronym for Sweat sampLIng DevicE, but it also represents the motion that is used for sampling—sliding on the surface of the skin of the armpit. Before extraction, the disk was cleaned with 80% isopropanol solution in a sonication bath for 30 min and then dried in air afterwards. After this procedure, the disc was ready to use for sweat sampling. After assembly, the glass filter projects out approximately 2 mm from the attachment.

The day before the collection, the participants shaved both axillae (dry or wet shaving procedure according to their habits) and afterwards they were not allowed to use any cosmetic products with underarm application until after the sweat collection procedure. Washing with soap or body shampoo when showering or bathing the evening before the collection was allowed. The next morning, the axillary apocrine sweat was collected as follows: one hour prior to sampling, the armpits were cleaned with distilled water and lightly wiped with a paper towel to remove surface dirt, microbial debris, dead cells and sebum, then the filter disk was placed on the attachment and it was carefully immersed in 80% isopropanol (so that it just touched the surface). The axillary apocrine sweat was then removed by passing the disk over the surface of the skin of the shaved part of the axilla, where the apocrine sweat glands open into the hair follicles. Apocrine sweat was expressed by slight pressure on the disk and immediately drawn by capillary forces into the interior of the disk. The extraction was performed bilaterally, with an estimated extraction time of approximately half a minute per axilla. One disk was always used for sampling only one side (left or right axilla). After the collection, the disk was pushed out of the attachment with tweezers (from the other side of the attachment) and transferred to a premarked tube, which was then immediately frozen at −80 °C until further sample preparation.

To test the reproducibility of this novel sampling approach, apocrine sweat was collected from 10 healthy individuals (A–J) in three consecutive sessions (1–3) from both the left (L) and right (R) sides providing six samples from each individual and a total of 60 samples (further description is provided in Appendix A).

### 4.4. Sample Preparation

Sweat samples were extracted from the porous glass filter disk by one-step isopropanol (IPA) extraction in random order. First, 400 µL of 80% IPA was added to the tube containing the disk and the mixture was sonicated for 10 min and incubated for 1 h at room temperature on a shaker. The extracts (200 µL) were transferred to Eppendorf tubes (1.5 mL), centrifuged (4 °C, 10,000× *g*, 10 min), and finally pipetted to glass vials for LC/MS analysis. An aliquot of 15 μL was taken and pooled from each extracted sample to make the final quality control (QC) sample. To eliminate potential systematic errors, QC were analyzed every 5th sample (for instrument stability control). Both orders of sample preparation and analysis were strictly randomized (for batch layout see Appendix A).

### 4.5. Pseudotargeted Lipidomic Analysis

The method for targeted lipidomic analysis, using liquid chromatography coupled to mass spectrometry, was adopted from Xuan et al. [43]. The LC separation was performed on ExionLC™ System (AB Sciex LLC, Framingham, MA, USA), the data were acquired using QTRAP^®^ 6500+ mass spectrometer (AB Sciex LLC, Framingham, MA, USA) and the system was controlled by Analyst software (version 1.6.2, AB Sciex LLC, Framingham, MA, USA). A reversed-phase BEH C8 (2.1 mm, 100 mm, 1.7 µm) column (Waters corporation, Milford, MA, USA) was used for the chromatographic separation. Mobile phase A consisted of ACN: H_2_O (6:4, *v/v*), the mobile phase B was IPA: ACN (9:1, *v/v*), and both contained 10 mM ammonium acetate. The flow rate was 0.35 mL/min and the column was tempered at 55 °C. The elution gradient was set as follows: commencement with 32% B for 1.5 min; linear increase to 85% B at 15.5 min; increase to 97% B at 15.6 min maintained for 2.4 min. The gradient then decreased to the initial composition of 32% B at 18.1 min and this ratio was maintained for 1.9 min for column equilibration.

The parameters of the ion source and gases of the mass spectrometer were set as follows: ion spray voltage +4500 V and −4500 V; curtain gas, 40 psi; ion source gases 1 and 2, 60 and 50 psi respectively; source temperature 400 °C. Scheduled MRM mode with a window of 2 min was applied for the data acquisition. Positive and negative ionization of compounds in one analysis was performed, using the polarity-switching feature of the used mass analyzer.

The adduct and fragmentation parameters (Appendix A) as well as the processing workflow were kept the same as in the original method. There was manual filtering of all theoretical MRM transitions (3800 ion pairs in two positive and one negative methods) based on multiple QC sample measurements (only peaks above three times the signal-to-noise ratio were selected), and correct identification was further verified by lipid pattern plots (plotted via the R script created by Drotleff et al. [44] and shown on Appendix A). QC samples were used for injection volume optimization (Appendix A), relevant MRM selection (the MRM parameters for 240 selected lipids are provided in Appendix A), and instrument stability measurement (Appendix A), and they were also used for locally estimated scatterplot smoothing (LOESS; Appendix A). Injection volume experiments were measured to verify a linear response for all analytes by injecting 0.5, 1, 1.5, 2, and 2.5 μL of a QC sample (Appendix A). Post-column injection of 100× diluted SPLASH^®^ LIPIDOMIX^®^ Mass Spec Standard mixture (with the syringe flow rate set at 3 µL/min) simultaneously during the analysis of QC samples was carried out for the evaluation of matrix effects. Non-scheduled MRM transitions corresponding to each deuterated standard in the mixture were measured to investigate ion suppression or ion enhancement across the time of analysis (8 in positive and 5 in negative mode and the dwell time was set to 30 ms) (Appendix A). To uncover potential interfering contaminants from the 3D-filament, 30s and 5 min extractions of this material were performed in 80% IPA (the top of the attachment was dipped in 1 mL of the extraction solution in a 5 mL glass beaker on a shaker) (Appendix A).

### 4.6. Data Treatment and Statistical Analysis

Data from the lipidomic analysis were integrated by SCIEX OS software (version 1.6.1, AB Sciex LLC, Framingham, MA, USA) and processed in the R program (version 4.0.3, R Foundation for Statistical Computing, Vienna, Austria) using the Metabol package [45]. Peak areas of analytes present in the process blank sample were subtracted from the areas in samples before further processing. Due to the variability of this biological material (discussed in Section 2.3), analyte peak areas in the blank were higher than in samples in a few cases. If this situation occurred in more than five samples, the whole analyte was discarded, otherwise values were set to half of a minimum value of a given analyte as described previously [46]. The QC-based, locally estimated smoothing signal correction (LOESS) was applied to the dataset, and coefficients of variation (CV) for all individual analytes were calculated from the QC samples. Metabolites with a CV higher than 30% were excluded from further data processing.

Data were statistically evaluated in GraphPad (version 9.0, San Diego, CA, USA) and SIMCA software (version 15.0, Umetrics, Umeå, Sweden). Ln and lnPQN transformation, Pareto scaling, and mean centring were applied to the final dataset. Data were evaluated by both multivariate (principal component analysis—PCA, and hierarchical cluster analysis—HCA) and univariate (*t*-test, box plots, violin plots) methods. The *p*-value was calculated by parametric *t*-test. The fold-change value was expressed as log two of the difference of medians of the given parameter of comparison. The Cytoscape program (version 3.8.2, https://www.cytoscape.org/, accessed on 1 July 2021) was used for global visualization of changes occurring in lipid profiles [25]. Each of the detected compounds is represented by a circle and significant metabolites/lipids are highlighted. The size of the circle represents the *p*-value (in −log value) and the depth of color is based on the fold-change (shades of red/blue represent an increase/decrease between two tested groups) or the % of relative variability.

### 4.7. Quantitative Evaluation of Lipid Profiles

A matrix-matched calibration series (created by dilution of the standard mixture by the QC sample) was prepared using isotopically labeled SPLASH^®^ LIPIDOMIX^®^ Mass Spec Standard mixture with the addition of two more deuterated standards: arachidonic acid (d8) and ceramide (d18:1d7/15:0). A 6-point calibration curve was plotted for each standard representing the whole lipid class and the corresponding equations were used for calculating concentrations (Appendix A). Values that were out of the range of the calibration curve were ignored for further processing. An isotopic correction factor for type I error was calculated and used to adjust the difference of ^13^C-abundance [28], while type II error was not considered to be isotopologues did not coelute in our RPLC separation setup (Appendix A).

## Figures and Tables

**Figure 1 ijms-22-08054-f001:**
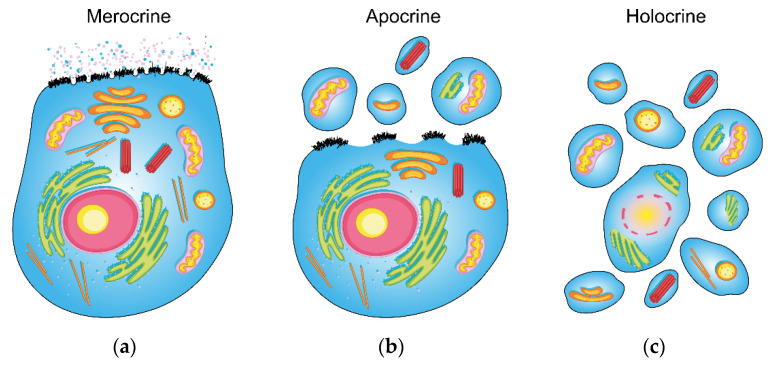
Schematic visualization of the three main types of sweat secretion (the design was adopted from Graphics RF, www.vecteezy.com, accessed on 1 July 2021). The general epithelial cell (with apical membrane) was chosen for the demonstration of: (**a**) merocrine, (**b**) apocrine and (**c**) holocrine secretion (a more detailed description of the mechanisms is provided in the introduction).

**Figure 2 ijms-22-08054-f002:**
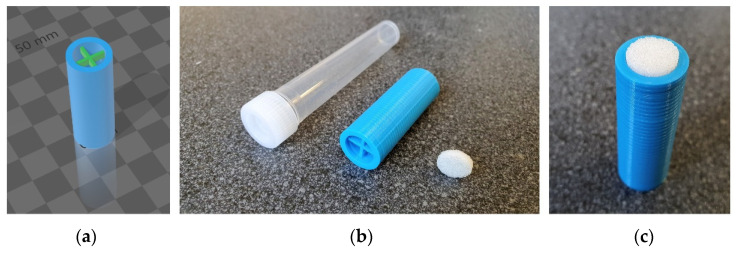
SLIDE—sweat sampling device consisting of a 3D-printed attachment, porous glass filter disk, and a plastic tube with a cap. 3D model of adapter (**a**), deconstructed parts (**b**), finally composed (**c**).

**Figure 3 ijms-22-08054-f003:**
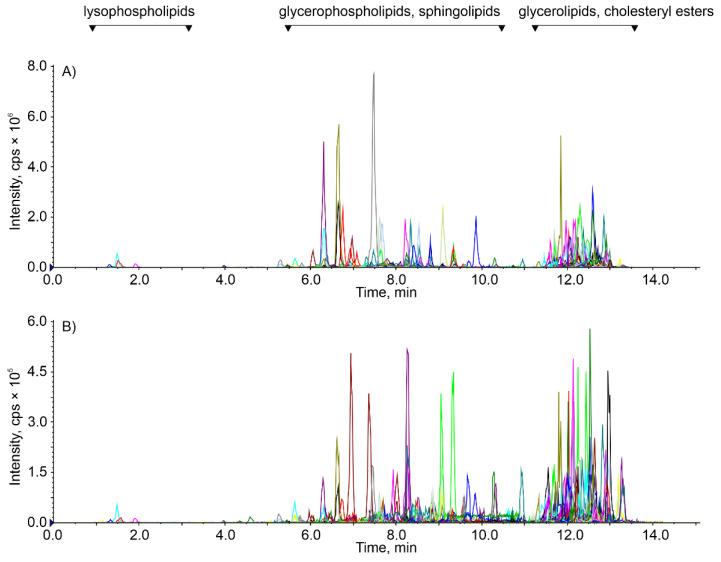
Chromatograms of samples D1R (**A**) and G3L (**B**) with different lipid profiles measured in positive ion mode. Each peak represents a selected MRM transition corresponding to a unique lipid.

**Figure 4 ijms-22-08054-f004:**
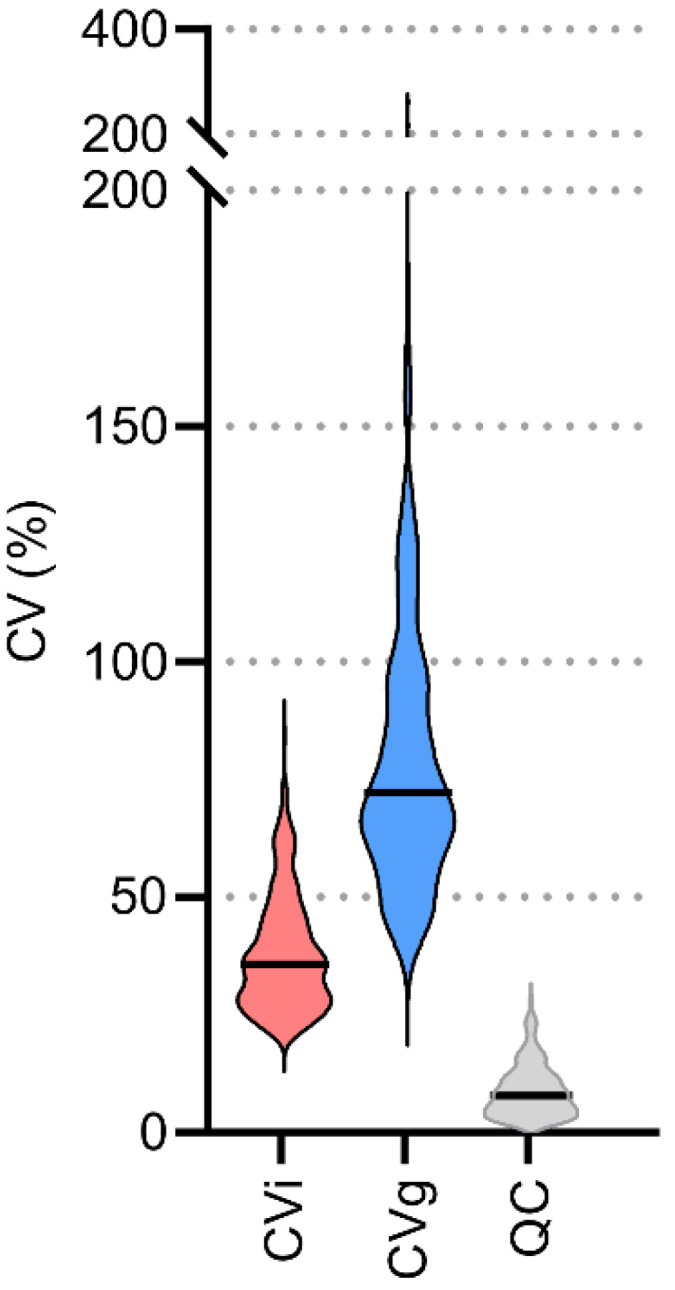
Overview of the variability (CV, %) for 240 detected lipids after lnPQN (natural logarithm with probabilistic quotient normalization) transformation. Quality control samples (QC), intraindividual (CVi), and group variability (CVg). Black lines: medians.

**Figure 5 ijms-22-08054-f005:**
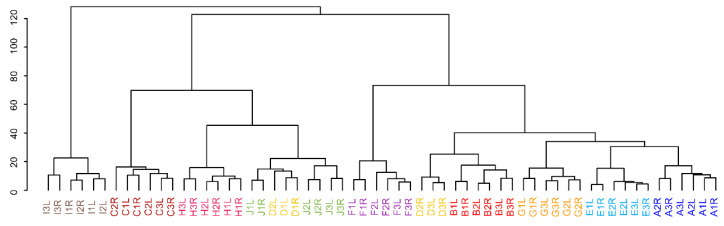
Hierarchical cluster analysis of lnPQN transformed data. Colors represent 10 individuals (A–J) who were sampled from the left (L) and right (R) sides for 3 different days (1–3).

**Figure 6 ijms-22-08054-f006:**
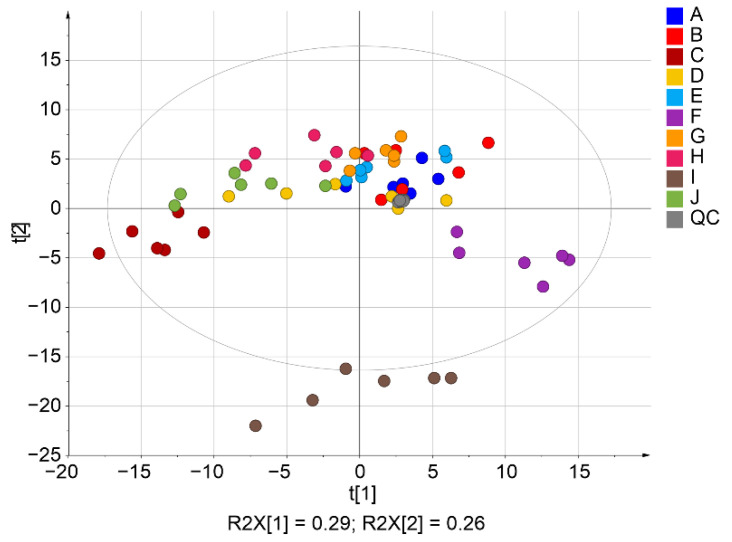
Principal component analysis of the lnPQN transformed data. Different colors represent 10 individuals (A–J).

**Figure 7 ijms-22-08054-f007:**
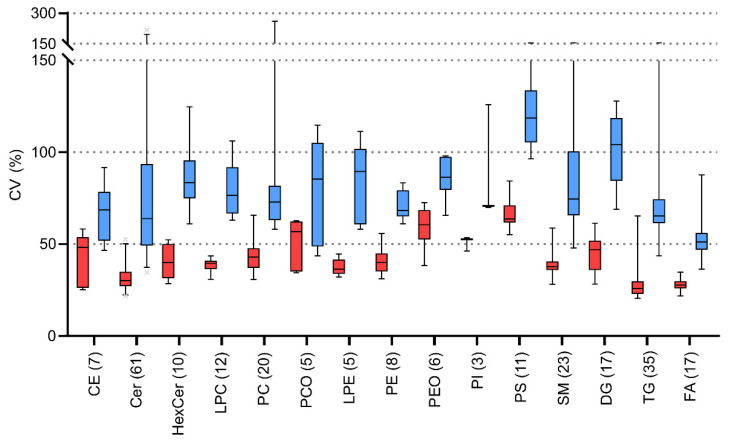
Boxplots of intraindividual (CVi, red boxes) and group variability (CVg, blue boxes) for detected lipids divided by lipid classes after application of lnPQN transformation. Boxes represent 1st and 3rd quartiles with centreline as the median. Lipid classes are abbreviated according to: cholesteryl esters (CE); ceramides (Cer); hexosylceramides (HexCer); lysophosphatidylcholines (LPC); phosphatidylcholines (PC); plasmalogen phosphatidylcholines (PCO); lysophosphatidylethanolamines (LPE); phosphatidylethanolamines (PE); plasmalogen phosphatidylethanolamines (PEO); phosphatidylinositols (PI); phosphatidylserines (PS); sphingomyelins (SM); diacylglycerols (DG); triacylglycerols (TG); free fatty acids (FA). Numbers in parentheses represent the number of identified lipids in each class.

**Figure 8 ijms-22-08054-f008:**
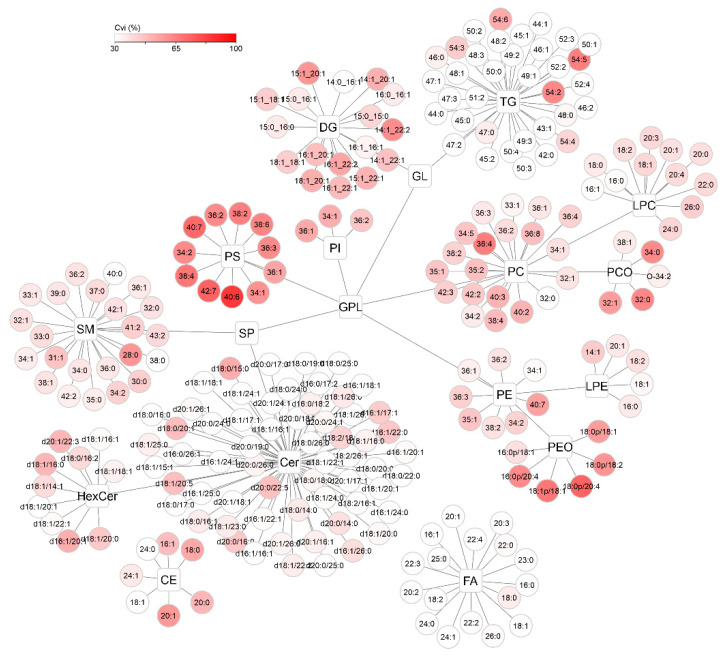
Cytoscape visualization of intraindividual variability (CVi, %) for all detected lipids after applying lnPQN transformation. The depth of color represents increasing relative variability (%).

**Figure 9 ijms-22-08054-f009:**
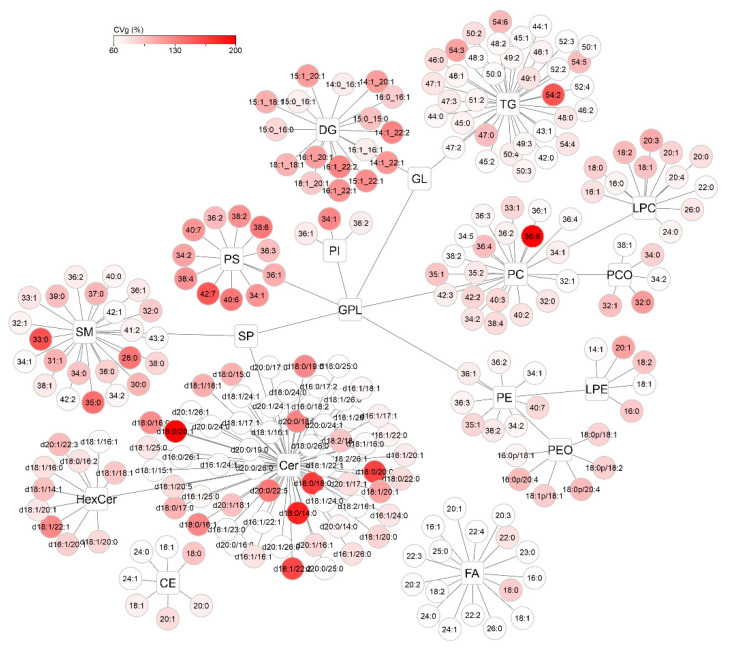
Cytoscape visualization of group variability (CVg, %) for all detected lipids after applying lnPQN transformation. The depth of color represents increasing relative variability (%).

**Figure 10 ijms-22-08054-f010:**
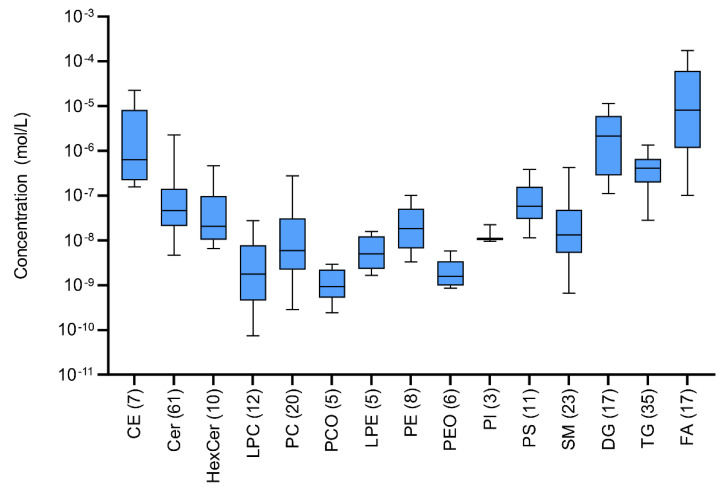
Boxplots of sweat extract concentration levels for quantified lipids in classes after the application of lnPQN transformation. Numbers in parentheses represent the number of identified lipids in each class.

**Figure 11 ijms-22-08054-f011:**
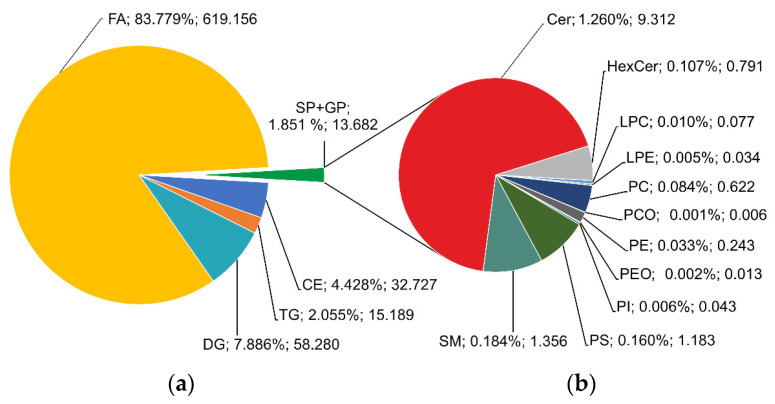
Total (**a**) and sectional (**b**) pie charts of summed relative content and average concentrations of lipids in classes (in % and µmol/L).

**Figure 12 ijms-22-08054-f012:**
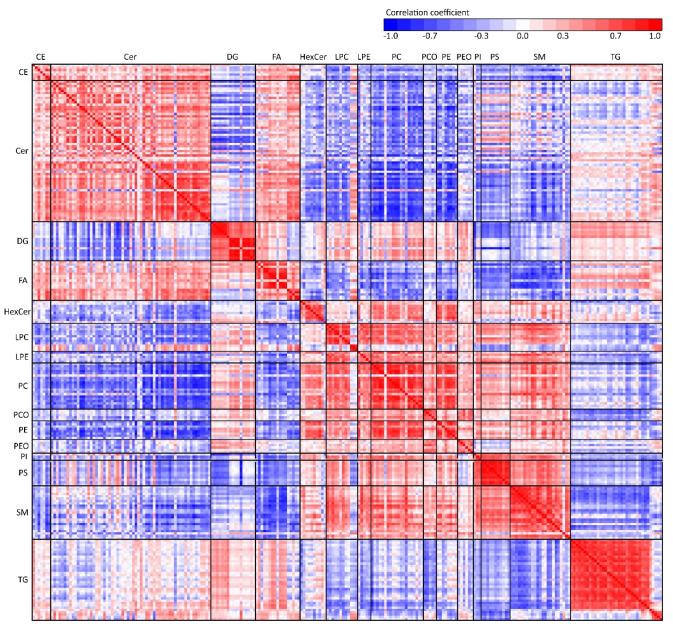
Heatmap of Pearson’s linear correlation coefficients between all lipid species categorized by lipid classes.

**Figure 13 ijms-22-08054-f013:**
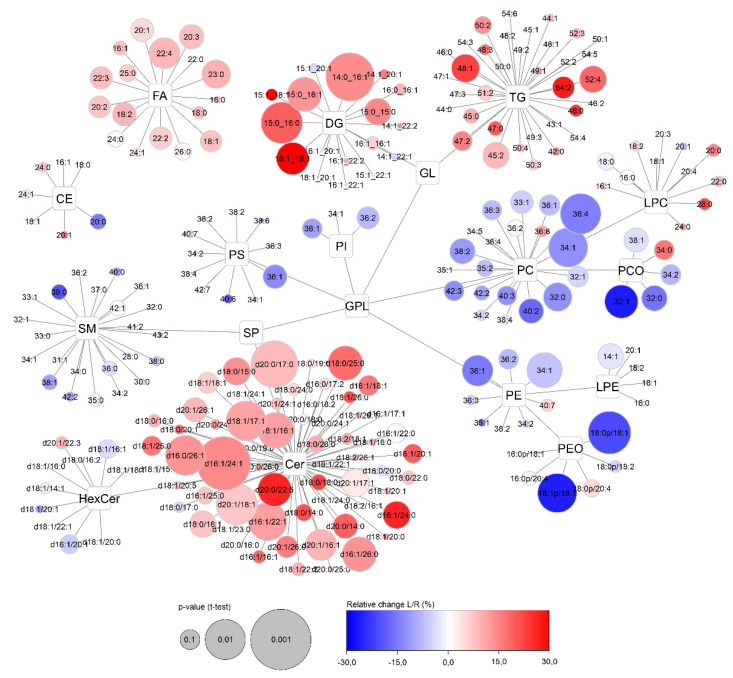
Overview of side-specific changes (left vs. right) in the apocrine sweat lipidome across all lipid classes. Color of nodes represents the relative ratio of the left and right side of sampling—L/R (in %, blue—decreased and red—increased) and their size depends on the *p*-value (paired *t*-test). Labels in individual lipid nodes describe the numbers of carbons and double bonds characterising the acyl-chain composition of the particular lipid.

## Data Availability

All data generated in this study are included in this published article and its Appendix A. The raw data files were deposited to the MassIVE database (ID: MSV000087871; doi:10.25345/C5GJ8D) with public access.

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
