# Peer review of "SLIDE—Novel Approach to Apocrine Sweat Sampling for Lipid Profiling in Healthy Individuals"

_ijms, 2021, doi:10.3390/ijms22158054_

Round 1
Reviewer 1 Report
The authors report the development of new device to detect apocrine sweat. The method is new and the evaluation through the lipidomics approach is sound. I have few minor comments only.
Minor:
- The authors need to clarify the condition for the MRM condition of 240 lipids (as a supplementary material).
- The isolation of sample I is not an anomaly; rather it is a supporting information that lipidomics can distinguish the age of samples. This fact is better moved to the result part rather than the discussion.
- The good performance of lnPQN is obvious, and its emphasis is better reduced. The comparative figures 4-9 can be reduced to results of only lnPQN.
- The larger detection of FA is noticiable. This part should be more explained.
- Section 2.6 is repeated.
- The raw data must be deposited to a public repository (e.g. MetaboLights or Massiv).
Reviewer 2 Report
In this paper “SLIDE – Novel Approach to Apocrine Sweat Sampling for Lipid Profiling in Healthy Individuals”, the authors have designed a concept of 3D-printed attachment with porous glass filter discs – SLIDE (Sweat Sampling Device) for easy sampling of apocrine sweat. The reproducibility and robustness of this novel approach were validated by complex lipid profiles, using LC-MS techniques. The SLIDE application as a sampling technique of apocrine sweat could bring a promising alternative, including various possibilities in modern clinical practice. Authors have composed paper very well and I recommend this work to be published after minor revision in the IJMS Journal. However, I have few comments and I recommend the authors address them.
My comments are below.
- Did author compare any traditional sample collection method with this method?
- Authors can provide the expanded stacked chromatograms of the lipid classes in supplementary information for Figure 3.
- Expand the legend of figure 3 as given information is not clear. Each line or each peak ?
Reviewer 3 Report
This manuscript describes the lipid profile of sweat sampled in the armpits of individuals using a home-developed device and methodology. Interesting features are shown as contrasting with previous comparable work using other sampling methodologies. The analytical and statistical methodologies are intensive and in agreement with most recent and proved approaches. Both methodologies are extensively described too. Figures and tables are of quality and well explained in the text.
Only two issues are of my concern in this study, which I ask the authors to address:
1) How much does the filter project out the device where it is fitted? This comment has to do with the possibility that during sliding some skin cells could have been scrapped out and contribute to the lipid profile. Isopropanol is a strong extractant for lipids.
2) In regard to the previous comment, was the sampling tested for a skin without sweat (blanc)? I understand this is difficult to achieve, but it could be relevant for data reliability.
3) Point 2 of presumed advantages of the technique for clinical applications (line 135): not all the cell membranes have identical lipid composition. Are your referring to the membrane of the gland cells or in general sense?
